# Mid-Arm Muscle Circumference or Body Weight-Standardized Hand Grip Strength in the GLIM Superiorly Predicts Survival in Chinese Colorectal Cancer Patients

**DOI:** 10.3390/nu14235166

**Published:** 2022-12-05

**Authors:** Tiantian Wu, Hongxia Xu, Yuanlin Zou, Jiuwei Cui, Kedi Xu, Mingming Zhou, Pengxia Guo, Haoqing Cheng, Hanping Shi, Chunhua Song

**Affiliations:** 1Department of Epidemiology and Statistics, College of Public Health, Zhengzhou University, Zhengzhou 450001, China; 2Department of Clinical Nutrition, Daping Hospital, Army Medical University, Chongqing 400042, China; 3Cancer Center, The First Hospital of Jilin University, Changchun 130021, China; 4Department of Gastrointestinal Surgery, Beijing Shijitan Hospital, Capital Medical University, Beijing 100054, China; 5Department of Clinical Nutrition, Beijing Shijitan Hospital, Capital Medical University, Beijing 100054, China

**Keywords:** colorectal cancer, GLIM, malnutrition, muscle mass, survival

## Abstract

Our objective was to identify the optimal method to assess reduced muscle mass (RMM) using the Global Leadership Initiative on Malnutrition (GLIM) approach and investigate the roles of the GLIM approach in nutrition assessment and survival prediction in colorectal cancer (CRC) patients. During a median follow-up period of 4.2 (4.0, 4.4) years, a development cohort of 3612 CRC patients with a mean age of 64.09 ± 12.45 years was observed, as well as an external validation cohort of 875 CRC patients. Kaplan–Meier curves and multivariate Cox regression were adopted to analyze the association between GLIM-diagnosed malnutrition and the overall survival (OS) of CRC patients. A nomogram predicting individualized survival was constructed based on independent prognostic predictors. The concordance index, calibration curve, and decision curve were applied to appraise the discrimination, accuracy, and clinical efficacy of the nomogram, respectively. Patients diagnosed with severe malnutrition based on either the mid-arm muscle circumference (MAMC) or body weight-standardized hand grip strength (HGS/W) method had the highest mortality hazard ratio (HR, 1.51; 95% CI, 1.34–1.70; *p* < 0.001). GLIM-defined malnutrition was diagnosed in 47.6% of patients. Severe malnutrition was an independent mortality risk factor for OS (HR, 1.25; 95% CI, 1.10–1.42; *p* < 0.001). The GLIM nomogram showed good performance in predicting the survival of CRC patients and was clinically beneficial. Our findings support the effectiveness of GLIM in diagnosing malnutrition and predicting OS in CRC patients.

## 1. Introduction

As the third most frequently diagnosed cancer and the second most deadly malignancy, colorectal cancer (CRC) induced almost 1.9 million new incidences and 0.9 million deaths worldwide in 2020 [1]. The situation is dire in China, where the incidence and mortality of CRC have steadily increased under the cancer transition stage resulting from socioeconomic development [2]. Although diverse advanced treatment strategies, including modified surgical technologies and neoadjuvant treatment, have promoted favorable results in the management of CRC, overall survival remains poor [3]. One of the determinants is malnutrition, which generally exists in CRC patients due to the tumor, diminished food intake or assimilation, and chronic blood loss [4].

Currently, a gold standard tool with which to diagnose malnutrition is absent [5]. According to different methods applied in the nutritional status appraisal, the incidence rate of malnutrition among CRC patients ranges from 31% [6] to 54% [7]. Moreover, malnutrition reduces the tolerability and effectiveness of various cancer treatments and is a capable predictor of surgical complications, prolonged hospital stays, readmission, and mortality among CRC patients [8,9]. To address these issues, it is imperative to add systematic nutrition support therapy into the conventional management of CRC. Therefore, an accurate malnutrition identification and classification system, as well as effective strategies for prevention and treatment, are urgently needed and of great significance for fundamental studies and clinical application.

In the recent past, the Global Leadership Initiative on Malnutrition (GLIM) proposed a two-step procedure for assessing nutritional status followed by three phenotypic and two etiologic criteria [10]. A large body of evidence has proved the validity of the GLIM process in distinguishing malnutrition [11,12,13]. There was an article demonstrating the ability of the GLIM process in nutritional status discrimination and survival prediction among older patients with malignant tumors [14]. Nonetheless, the conclusion of older adult cancer patients was not applicable to CRC patients. The effect of the GLIM approach on discriminating malnutrition and the degree of severity, as well as predicting OS among CRC patients, remains to be determined. Additionally, as one of the phenotypic criteria in the GLIM process, the unified evaluation metrics of reduced muscle mass (RMM) were absent, and the optimal method to assess RMM using the GLIM approach for CRC patients has yet to be explored. Several anthropometric indicators, including mid-arm circumference (MAC), calf circumference (CC), and hand grip strength (HGS), have been suggested as alternative measurements to appraise RMM. To meet the needs of precision medicine and better clinical work, a further study exploring the best RMM evaluation method is an imperative challenge. Given this, this study was performed to identify the optimum combination of diverse anthropometric parameters to evaluate RMM and explore the utilization of the GLIM criteria in nutritional status assessment and survival prognostication in CRC patients.

## 2. Materials and Methods

### 2.1. Study Design and Population

This study was conducted based on the Investigation on Nutrition Status and its Clinical Outcome of Common Cancers (INSCOC) project from China (chictr.org.cn (accessed on 1 October 2022): ChiCTR1800020329), which is a multicenter prospective cohort study, having recruited more than 60,000 patients covering 18 types of cancers across over 100 institutions from July 2013 to March 2022. The study design of INSCOC has been described in detail [15]. The inclusion and exclusion criteria of the INSCOC project are recorded in Appendix A. Data on colorectal cancer (CRC) patients (10,214) from the INSCOC project were used for the present study. After ruling out 1791 patients with incomplete basal information and 5403 patients lost or refusing to follow-up, a total of 3612 CRC patients finally served as our development cohort, which was used for selecting optimal muscle mass indices to assess RMM with the GLIM standard and the development of the model (nomogram) with internal validation. The median time at follow-up was 1540 (1478, 1624) days. In addition, two independent cohorts, totaling 875 patients enrolled from 2 centers, served as the external validation cohort, with patients from the First Affiliated Hospital of Kunming Medical University included between May 2013 and April 2021 (*n* = 410) and the First Affiliated Hospital of Sun Yat-sen University between January 2012 and July 2015 (*n* = 465). All patients in the external validation cohort conformed with the inclusion and exclusion criteria of the development cohort (Appendix A). This study was approved by the Ethical Review Board of each participant hospital, and was performed with the written informed consent of patients and in accordance with the Declaration of Helsinki.

### 2.2. Data Acquisition

Within the first 48 h of admission, the following preoperative information was collected by a project-trained nutritionist or clinician for each patient: recent nutrition information via a comprehensive conversation with patients; the Nutrition Risk Screening 2002 (NRS 2002) score, the Patient-Generated Subjective Global Assessment (PG-SGA); the Karnofsky performance score (KPS); and laboratory tests.

The database also captured information on demographic and tumor-related variables, ranging from age to gender, family history, lifestyle, TNM stage, comorbidity, type and duration of cancer treatment, and so forth. Follow-up was carried out annually to attain the survival condition.

### 2.3. Anthropometric Measurements

The project-trained nutritionist or clinician was instructed to take the measurement in person, avoid asking the patient, and fill it in directly. Height and weight measurement require an empty stomach, no shoes, and single clothes. Body mass index (BMI) was obtained by dividing weight (kg) by height (m) squared. The mid-arm circumference (MAC) and triceps skinfold thickness (TSF) were examined using the nondominant arm, usually the left arm, and with the patient’s arms held naturally drooping. MAMC was computed using the equation MAC (cm) − 3.14 × TSF (cm). Hand grip strength (HGS) was evaluated with the nondominant arm, and the average value was obtained by measuring it three times. The body weight-standardized HGS (HGS/W) was obtained by dividing the HGS by the weight. When measuring the calf circumference (CC), the patient was required to lie on their back. The left leg would be measured three consecutive times, and the maximum value would be taken.

### 2.4. Nutrition Status Assessment

The GLIM consensus recommends a 2-step method, including risk screening and diagnosis [10]. The initial screening was simultaneously implemented with two validated screening tools: the NRS-2002 and PG-SGA. Patients with NRS 2002 scores ≥ 3 or PG-SGA scores ≥ 4 were regarded as being at risk of malnutrition. Subsequently, participants meeting a combination of at least one phenotypic criterion (weight loss, body mass index, reduced muscle mass) and one etiologic criterion (reduced food intake or assimilation, inflammation) were identified as malnourished, since CRC could definitely affect food intake or absorption, so patients satisfying one of three phenotypic criteria were recognized as malnourished in our research. Next, severity grading was only determined with the phenotypic criterion.

Concerning phenotypic criteria, unintentional weight loss was estimated by comparing the historical weight within six months with the weight measured upon admission (formula: [historical weight − measured weight]/historical weight), and reference values for severity grading were noted in the GLIM consensus, which also documented the Asian criteria for low BMI. Moreover, severity grading was judged on the basis of cut-off values recommended by a former study carried out among the Asian population (17.0 kg/m^2^ for patients aged < 70 years and 17.8 kg/m^2^ for patients aged ≥ 70 years) [16]. RMM was appraised from two aspects: muscle mass was estimated with the MAMC and CC, while a muscle function assessment was established with HGS/W, which has been proven to be more accurate in predicting prognosis compared with HGS alone [17]. The 5th percentile (p5) and 15th percentile (p15) of the MAMC, CC, and HGS/W were computed independently for males and females. A value < p15 was considered indicative of stage I/moderate malnutrition, and <p5 was classified as stage II/severe malnutrition. Nine diverse combinations of MAMC, CC, and HGS/W were applied to evaluate RMM and establish the GLIM severity grading, including (1) MAMC positive, (2) CC positive, (3) HGS/W positive, (4) either MAMC or HGS/W positive, (5) both MAMC and HGS/W positive, (6) either CC or HGS/W positive, (7) both CC and HGS/W positive, (8) MAMC or CC or HGS/W positive, and (9) MAMC and CC and HGS/W all positive. The combination that most accurately predicted survival probability was singled out for subsequent analysis.

### 2.5. Selection of Predictors Associated with the OS

A univariate Cox regression analysis was performed with the development cohort for initial screening of all characteristics (*p* < 0.1). Moreover, the least absolute shrinkage and selection operator (LASSO) method was utilized for further dimension shrinking [18]. The chosen predictors were converged for multivariate Cox regression analysis. A dual-direction stepwise procedure based on the Akaike information criterion statistic was adopted to identify the most significant predictors for inclusion in the nomogram.

### 2.6. Development and Validation of a Nomogram for Survival Prediction

Independent prediction factors filtered by the multivariate Cox regression analysis (*p* < 0.05) were incorporated to construct a nomogram to predict individualized survival. The concordance index (C-index) with internal bootstrap correction (1000 repetitions), as well as time-dependent C-index curves, were calculated and plotted to quantify the discrimination capability of the nomogram [19]. A calibration curve analysis based on the bootstrapping method was conducted to appraise the prediction accuracy of the nomogram by comparing survival probability between the nomogram-predicted values and the actually observed values [20]. To check for the generalizability of the model in CRC patients, discrimination and calibration were also checked in the external validation cohort [21].

### 2.7. Clinical Application

Decision-curve analysis was performed in the development cohort to assess the clinical utility of the nomogram by calculating and plotting the net benefits at different threshold probabilities of death during follow-up [22]. Moreover, to compare the clinical efficiency between the nomogram and TNM classification, a decision curve was also plotted for the model comprising only the TNM stage [23].

### 2.8. Statistical Analysis

Continuous variables were presented as means ± standard deviation and compared utilizing *t*-tests. The normality of data was examined with a Shapiro–Wilk test. Categorical variables were presented as numbers (percentages) and compared using χ^2^ tests. Kaplan–Meier curves and Cox regressions were applied to process survival data. All tests were two-tailed and *p* < 0.05 was considered statistically significant. All analyses were carried out with software R 4.2.0, Vienna, Austria (http://www.rproject.org (accessed on 1 September 2022)).

## 3. Results

### 3.1. MAMC or HGS/W Method Has Optimal Performance for the Evaluation of RMM

After applying different RMM assessment approaches to the GLIM diagnosis, less than half of patients with normal nutritional status died; thus, no median survival time was observed with the MAMC or HGS/W (either positive), CC or HGS/W (either positive), and MAMC or CC or HGS/W (any positive) methods. Of the three methods, patients with severe malnutrition diagnosed with the MAMC or HGS/W method had the shortest median survival (1176 days; 95%CI, 1074–1328) (Appendix A, Table 1). Simultaneously, patients with severe malnutrition diagnosed using the MAMC or HGS/W method had the highest mortality hazard ratio (HR) compared with patients in the normal group (HR, 1.51; 95% CI, 1.34–1.70; *p* < 0.001) (Table 2). Taken together, using the MAMC or HGS/W method to evaluate RMM had the optimal performance in identifying survival-related malnutrition, and this approach was selected for the following analyses. Moreover, the results shown in Table 2 indicated that the predictive effects of different RMM assessment approaches on an increased mortality risk are presented in CRC patients with moderate and severe malnutrition compared to normal nutritional status.

### 3.2. Baseline Characteristics

The parameters and threshold values of the GLIM standard are exhibited in Table 3. GLIM-stratified baseline characteristics of the development cohort are presented in Table 4, and more detailed demographic features and clinical baseline characteristics of the development and validation cohort are provided in Appendix A. There were 2173 males and 1439 females with a mean age of 64.09 ± 12.45 years. A total of 2394 (66.3%) CRC patients in the development cohort were regarded as being at risk of malnutrition based on the NRS 2002 and PG-SGA scores. According to the GLIM diagnosis and severity classifying criteria (based on the MAMC or HGS/W method), 1720 (47.6%) CRC patients were malnourished, of which 26.2% and 21.4% were rated as moderate and severe malnutrition, respectively. There were no statistical discrepancies in gender distribution, smoking status, drinking status, family history, TNM stage, organ metastasis, and radical resection status of CRC patients among the three groups. The GLIM-diagnosed malnutrition was positively related to age, NRS 2002 and PG-SGA scores, direct bilirubin and C-reactive protein levels, and counts of white blood cells and neutrophils (all *p* < 0.05). On the contrary, there was a negative association between GLIM-diagnosed malnutrition and differentiation degree, adjuvant chemotherapy, KPS score, BMI, MAMC, HGS/W, CC, total protein, albumin, prealbumin, hemoglobin, red blood cell and platelets counts (all *p* < 0.05). All consequences demonstrated that GLIM diagnostic criteria could indeed reflect the nutritional status in CRC patients. For the validation cohort, 473 (54.1%) patients were malnourished based on the GLIM criteria, and higher rates of moderate (27.7%) and severe malnutrition (26.4%) were observed compared to the development cohort.

### 3.3. Kaplan–Meier Analysis

According to the Kaplan–Meier curves, patients in the moderate and severe malnutrition group obviously had shorter overall survival (OS) than those in the normal group (median OS: normal = not reached, malnutrition = 1176 days, *p* < 0.0001, Figure 1A). Additionally, the correlation between the GLIM severity and the OS in the development cohort was negative (median OS: normal = not reached, stage I = 1549 days, stage II = 1176 days, *p* < 0.0001, Figure 1B).

### 3.4. Predictors Associated with Survival

After initial screening with univariate Cox regression analysis, 96 baseline characteristics were reduced to 62 variables, from which LASSO regression analysis selected 7 predictors for multivariate Cox regression analysis (λ = 0.057) (Appendix A). With a dual-direction stepwise regression screening, all seven predictors were significant: TNM stage, organ metastasis, KPS, prealbumin, direct bilirubin, hemoglobin, and GLIM-diagnosed malnutrition.

### 3.5. GLIM-Diagnosed Malnutrition as an Independent Mortality Risk Factor for Survival

A forest plot shows the results of the final model (Figure 2). An analysis of the adjusted multivariate Cox regression indicated that an advanced tumor stage, organ metastasis, abnormal prealbumin (<0.28 g/L), hemoglobin (males < 120 g/L, females < 110 g/L), direct bilirubin levels (>6.8 μmol/L), and severe malnutrition were independent mortality risk factors for survival (all *p* < 0.05). Conversely, the KPS score was an independent protective factor for survival (*p* < 0.05). Notably, after multivariable adjustment, severe malnutrition remained an independent prognostic factor for survival in CRC patients and resulted in a 1.25 times mortality risk compared with the normal nutritional status (HR, 1.25; 95% CI, 1.10–1.42; *p* < 0.001).

### 3.6. Nomogram and Its Performance

Seven independent predictors were integrated into a nomogram model, which was used to predict the patients’ survival (Figure 3). Both the calibration curves in the development and validation cohort depicted good agreement between predicted and observed values in the probability of 3-year survival (Figure 4A,B). In addition, corrected C-indexes (95% CIs) for the development and validation cohort were 0.733 (0.721–0.745) and 0.640 (0.613–0.667), respectively. The time-dependent C-index is exhibited in Figure 4C,D. To sum up, the GLIM nomogram evidently had a good discrimination and calibration capability.

### 3.7. Clinical Utilization of the Nomogram

Three decision curves at 1, 3, and 5 years consistently showed greater benefits for the GLIM nomogram than the classical American Joint Committee on Cancer TNM staging system at all threshold probability values, as shown in Figure 4E.

## 4. Discussion

Malnutrition is not only a global public health challenge, but also a prevailing clinical condition [24], linked in particular to detrimental consequences in cancer patients [25]. The universality of malnutrition among CRC patients has been demonstrated in numerous studies [26]. Thus, a globally accepted diagnostic criterion for malnutrition is imperative to normalize the nutrition assessment and management of CRC patients. Recently, the GLIM consensus recommended an updated approach to identify nutritional status on the bases of phenotypic and etiologic criteria, and, soon after, plentiful research proved the validity of this consensus in diagnosing malnutrition in specific populations, such as those with chronic obstructive pulmonary disease [27], ICU patients [28], and so on. However, the best assessment method for RMM, which is one of the phenotypic criteria in the GLIM approach, and the effectiveness of the GLIM criteria in CRC patients have yet to be investigated. In the present research, we observed that the condition of either MAMC or HGS/W being positive for assessing RMM was the finest method to identify malnutrition linked to survival. The incidence of malnutrition was 47.6%; among the malnourished patients, moderate malnutrition accounted for 26.2% and severe malnutrition accounted for 21.4%. Moreover, severe malnutrition diagnosed using the GLIM standard correctly revealed poor nutritional condition and was an independent mortality risk factor for the survival of CRC patients. In addition, a GLIM-related nomogram constructed to predict patient survival showed good performance both in the development and validation cohort, and it was clinically useful.

Compared with a previous study that focused on the applicability of the GLIM standard among old cancer patients [14], we focused more precisely on CRC patients. A certain extent of bias exists when applying conclusions of elderly cancer patients to CRC patients. Therefore, we targeted CRC patients and found a similar ability of the GLIM approach in diagnosing malnutrition and predicting OS in old cancer patients. More importantly, we further explored the best assessment method for RMM in CRC patients, which was not investigated in that study. The combination of either MAMC or HGS/W being positive was optimal to evaluate RMM using the GLIM criteria. This finding was essential to clinical work and is in line with the development of precision medicine.

Patients with RMM are likely to develop sarcopenia, which is related to frailty, cachexia, and functional disability, resulting in a worse quality of life and elevated mortality rates [29]. Adding its prognostic value in cancer patients, the identification of RMM becomes increasingly relevant in the GLIM consensus. With regard to RMM evaluation, the combination of either MAMC or HGS/W being positive was found to be superior to other combinations of several indexes in discriminating malnutrition associated with survival. As previously mentioned, MAMC was measured using the values of TSF and MAC, which could stably reflect the subcutaneous adipose tissue and muscle mass of the mid-arm. The role of MAMC in distinguishing poor muscle mass in the clinical application was demonstrated in a comparison study [30]. Moreover, a prospective cohort study indicated that MAMC was inversely related to all-cause mortality, and this association was not affected by BMI [31]. Considering our results and previous evidence, the inclusion of MAMC in the optimal RMM assessment for GLIM criteria was reasonable.

Notably, the top five RMM evaluation methods in the current study all involved the HGS/W, which was considered a supportive measurement for muscle function by the GLIM [10]. This phenomenon also suggests that the HGS/W is an effective and vital parameter for RMM assessment, which was consistent with a previous study that indicated that HGS could provide significant prognostic information about complication and mortality risks for medical inpatients at nutritional risk, and help to recognize patients gaining most from medical nutrition treatment [32]. Moreover, a multicenter observational study proved the association between low HGS and a poor survival probability in one year of patients with cancer cachexia [33]. According to the latest research, HGS/W exhibited better accuracy in predicting a prognosis than using HGS alone [17]. To sum up, HGS/W appears to be a reliable and effective indicator of RMM evaluation in the GLIM criteria. Additionally, without cut-off values of MAMC and HGS/W in Chinese or Asian populations, we separately computed the 5th and 15th percentile of the development cohort patients for the different genders. In future research, these could be considered as reference values, and it would be better if the validation in various kinds of populations covering wider age distributions was conducted.

With a freshly obtained RMM assessment, the GLIM diagnoses were implemented among CRC patients. The incidence of malnutrition in the present research (47.6%) was at a moderate level compared with previous studies using different diagnostic methods [26,34]. Interestingly, a recent study also conducted the GLIM diagnoses in CRC patients, but with four different risk screening tools in the first step, leading to a prevalence of malnutrition ranging from 10% to 24% [5], which was much lower than our result. Except for the small sample, another reason for this is that they used four screening tools alone, instead of combining them, as in our research (either the NRS 2002 or the PG-SGA). Moreover, comparing various nutrition-related information between normal and malnourished patients, we found that malnourished patients were indeed under poor nutritional status, and severe malnutrition was an independent mortality risk factor for overall survival in CRC patients. To the best of our knowledge, this is the first study that has validated the effect of the GLIM criteria for identifying survival-related malnutrition in CRC patients, thus, providing supportive evidence for the utilization of the GLIM approach among CRC patients in future research.

Previous studies have disclosed considerable prognostic factors for CRC survival, such as age [35], chemotherapy [35], BMI [36], histology type [37], TNM stage [37], differentiation grade [38], C-reactive protein [39], total bile acid [40], direct bilirubin [40], neutrophils [41], and so forth. Therefore, we comprehensively collected the baseline characteristics of the development cohort. In a univariate Cox regression analysis, we found that elevated PG-SGA and NRS-2002 scores were significantly related to a shorter OS, and a higher BMI category was a protective factor for OS in CRC patients (Appendix A). However, as conventional predictors for malnutrition and prognosis, they were not distinguished as significant prognostic factors in the LASSO regression in this study. The most plausible explanation for this is that GLIM-diagnosed malnutrition, which reflects both the physical condition and illness status, was a more accurate and robust predictor of prognosis compared with individual factors. These discoveries affirmed the beneficial effect of the GLIM criteria for identifying survival-related malnutrition in CRC patients and were in conformity with previous studies conducted among lung cancer [42] and gastric cancer patients [43]. Based on the significant prognostic value of GLIM-diagnosed malnutrition, a novel GLIM nomogram was established for predicting overall survival in CRC patients. With good discrimination, a high accuracy was observed in the development and external validation cohorts and more significant benefit compared with the TNM staging system at all threshold probability values, so the GLIM nomogram may prove to be a helpful tool for clinical decision-making.

The application of bioelectrical impedance analysis (BIA) to measure body composition is an analytical technique that has been developed in the last 20 years. BIA can be used to measure body composition parameters such as the percentage of body fat and fat-free body mass index. However, only body composition data can be obtained through BIA; it is not comprehensive enough to assess the nutrition status without an evaluation of patients’ disease condition. With three phenotypic and two etiologic criteria, the GLIM criteria can comprehensively assess patients’ nutrition status and classify the malnutrition severity. Therefore, GLIM might be preferred over BIA in malnutrition diagnosis.

There are several strengths in our research. Firstly, this study was a multicenter prospective cohort study with a high level of evidence quality, and data were collected over 10 years of follow-up. Moreover, the sizable sample that was enrolled from multiple institutions increases the research reliability and representativeness. Furthermore, validation in an external cohort makes it possible to generalize the GLIM nomogram. Some potential limitations could not be ignored. First, postoperative complications were not considered in the research design; this might cause a confounding bias that confused the correlations we observed. Second, RMM was assessed based on anthropometry rather than CT scans, which might lead to a fraction of error in the classifications. To our knowledge, a cross-sectional CT image of the third lumbar spine is the gold standard for assessing patients’ muscle mass and diagnosing sarcopenia. However, CT scans require medical professionals and are expensive, so they are not universally available. In our cohort, cross-sectional CT images of the third lumbar spine were not performed, and the relevant data were not acquired, so we used anthropometric measurement parameters to evaluate muscle mass. In future, we will consider adding CT scans to assess the patients’ muscle mass and conduct a further study to compare the effect of CT and anthropometry indicators on assessing muscle mass. Third, we only considered the independent effect of CRP and albumin in the screening process, while the combined effect of two variables was ignored. The CRP/Albumin ratio has been indicated to be an effective prognostic predictor in previous studies [44,45]. Future studies focusing on the performance of CRP/Albumin ratio in predicting overall survival in Chinese cancer patients are warranted.

## 5. Conclusions

The combination of either MAMC or HGS/W being positive was optimal to evaluate RMM in the GLIM criteria. Moreover, severe malnutrition diagnosed by the GLIM approach was an independent mortality risk factor for survival in CRC patients. Furthermore, the GLIM nomogram is clinically beneficial for survival prediction.

## Figures and Tables

**Figure 1 nutrients-14-05166-f001:**
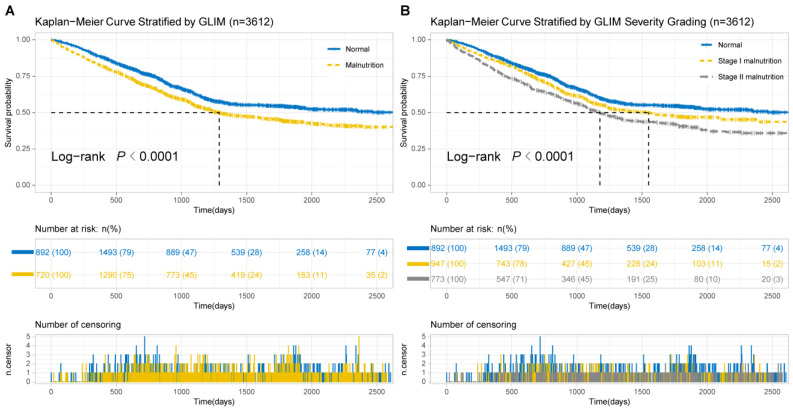
Kaplan–Meier curve analysis on the Global Leadership Initiative on Malnutrition (GLIM) grade. (**A**) Survival stratified by malnutrition. (**B**) Survival stratified by the severity of malnutrition. GLIM, the Global Leadership Initiative on Malnutrition.

**Figure 2 nutrients-14-05166-f002:**
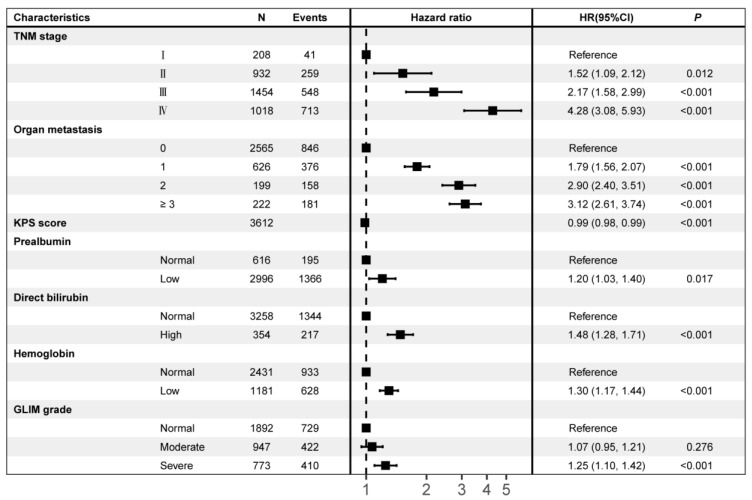
Multivariate Cox Regression Analysis of the Association Between the GLIM Grade and Overall Survival. HR, hazard ratio; CI, confidence interval; KPS, Karnofsky performance score; GLIM, the Global Leadership Initiative on Malnutrition.

**Figure 3 nutrients-14-05166-f003:**
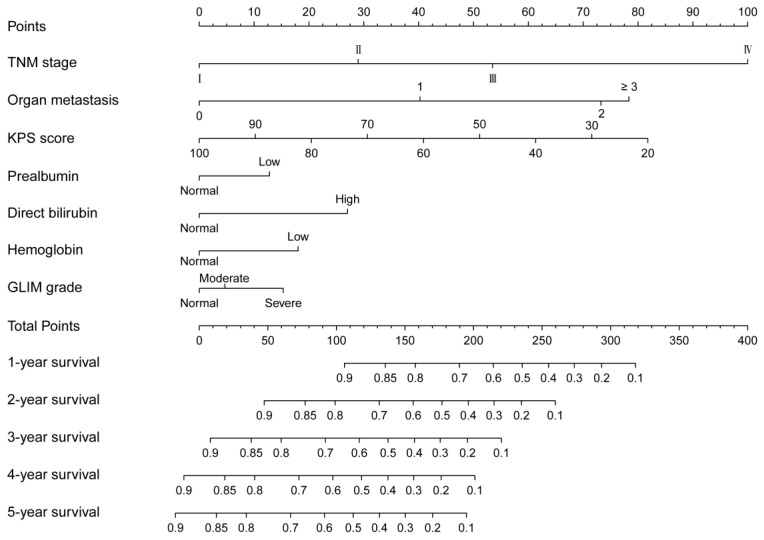
The nomogram for overall survival prediction in CRC patients. KPS, Karnofsky performance score; GLIM, the Global Leadership Initiative on Malnutrition.

**Figure 4 nutrients-14-05166-f004:**
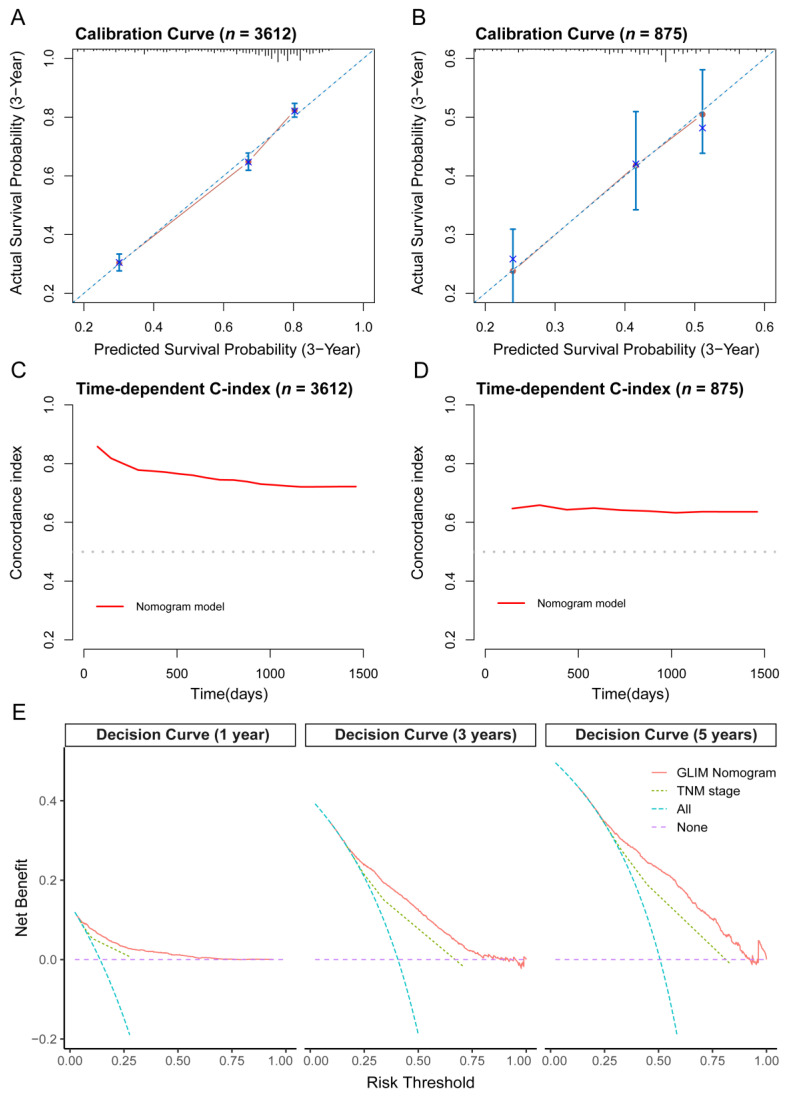
Validation of the nomogram for overall survival prediction in CRC patients. (**A**) The calibration curve for predicting patient survival at 3 years in the development cohort and (**B**) validation cohort. (**C**) Time-dependent C-index of the nomogram in the development cohort and (**D**) validation cohort. (**E**) The decision-curve analysis of the GLIM nomogram and TNM stage at 1, 3, and 5 years (in the development cohort). C-index, Concordance index; GLIM, the Global Leadership Initiative on Malnutrition.

**Table 1 nutrients-14-05166-t001:** Median Survival Days of the Study Population Stratified by GLIM Grade Using Different Parameters for RMM Assessment.

Parameters to Assess RMM	Normal		Stage I (Moderate Malnutrition)		Stage II (Severe Malnutrition)		*p*
	*n* (%)	Median (95% CI)	*n* (%)	Median (95% CI)	*n* (%)	Median (95% CI)	
No RMM assessment *	2140 (59.3)	2352 (1893, NA)	876 (24.3)	1455 (1230, 1770)	596 (16.5)	1176 (1054, 1412)	<0.001
MAMC	2027 (56.1)	2464 (1949, NA)	908 (25.1)	1373 (1224, 1754)	677 (18.7)	1180 (1076, 1353)	<0.001
CC	2051 (56.8)	2464 (1908, NA)	747 (20.7)	1589 (1255, 2118)	814 (22.5)	1180 (1074, 1350)	<0.001
HGS/W	1988 (55.0)	2464 (1948, NA)	930 (25.8)	1549 (1248, 2118)	694 (19.2)	1160 (1056, 1346)	<0.001
MAMC or HGS/W	1892 (52.4)	NA (1971, NA)	947 (26.2)	1549 (1241, 2086)	773 (21.4)	1176 (1074, 1328)	<0.001
MAMC and HGS/W	2123 (58.8)	2464 (1904, NA)	891 (24.7)	1373 (1224, 1754)	598 (16.6)	1176 (1056, 1373)	<0.001
CC or HGS/W	1912 (52.9)	NA (1949, NA)	807 (22.3)	1615 (1300, NA)	893 (24.7)	1180 (1076, 1328)	<0.001
CC and HGS/W	2127 (58.9)	2464 (1904, NA)	870 (24.1)	1455 (1231, 1833)	615 (17.0)	1160 (1047, 1350)	<0.001
MAMC or CC or HGS/W	1839 (50.9)	NA (1971, NA)	831 (23.0)	1615 (1296, NA)	942 (26.1)	1183 (1077, 1310)	<0.001
MAMC and CC and HGS/W	2136 (59.1)	2352 (1895, NA)	878 (24.3)	1408 (1206, 1766)	598 (16.6)	1176 (1056, 1373)	<0.001

GLIM, Global Leadership Initiative on Malnutrition; RMM, reduced muscle mass; CI, confidence interval; MAMC, mid-arm muscle circumference; CC, calf circumference; HGS/W, hand grip strength/weight; * for no RMM assessment, only weight loss and body mass index were considered as the phenotypic criteria for GLIM diagnosis implement.

**Table 2 nutrients-14-05166-t002:** Univariate Cox Regression Analyses of the GLIM Grade and Survival Using Different Parameters for RMM Assessment.

Parameters to Assess RMM	Normal		Stage I (Moderate Malnutrition)		*p*	Stage II (Severe Malnutrition)		*p*
	*n* (%)	Reference	*n* (%)	HR (95% CI)		*n* (%)	HR (95% CI)	
No RMM assessment *	2140 (59.3)	1	876 (24.3)	1.21 (1.07, 1.36)	0.002	596 (16.5)	1.44 (1.26, 1.64)	<0.001
MAMC	2027 (56.1)	1	908 (25.1)	1.24 (1.10, 1.40)	<0.001	677 (18.7)	1.45 (1.28, 1.65)	<0.001
CC	2051 (56.8)	1	747 (20.7)	1.18 (1.03, 1.34)	0.014	814 (22.5)	1.45 (1.29, 1.63)	<0.001
HGS/W	1988 (55.0)	1	930 (25.8)	1.19 (1.06, 1.34)	0.004	694 (19.2)	1.50 (1.32, 1.70)	<0.001
MAMC or HGS/W	1892 (52.4)	1	947 (26.2)	1.21 (1.07, 1.36)	0.002	773 (21.4)	1.51 (1.34, 1.70)	<0.001
MAMC and HGS/W	2123 (58.8)	1	891 (24.7)	1.23 (1.09, 1.38)	0.001	598 (16.6)	1.44 (1.27, 1.64)	<0.001
CC or HGS/W	1912 (52.9)	1	807 (22.3)	1.16 (1.02, 1.32)	0.026	893 (24.7)	1.48 (1.32, 1.67)	<0.001
CC and HGS/W	2127 (58.9)	1	870 (24.1)	1.21 (1.07, 1.36)	0.002	615 (17.0)	1.47 (1.29, 1.67)	<0.001
MAMC or CC or HGS/W	1839 (50.9)	1	831 (23.0)	1.18 (1.04, 1.34)	0.011	942 (26.1)	1.51 (1.35, 1.69)	<0.001
MAMC and CC and HGS/W	2136 (59.1)	1	878 (24.3)	1.22 (1.08, 1.37)	0.001	598 (16.6)	1.44 (1.26, 1.64)	<0.001

GLIM, Global Leadership Initiative on Malnutrition; RMM, reduced muscle mass; HR, hazard ratio; CI, confidence interval; MAMC, mid-arm muscle circumference; CC, calf circumference; HGS/W, hand grip strength/weight; * For no RMM assessment, only weight loss and body mass index were considered as the phenotypic criteria for GLIM diagnosis implement.

**Table 3 nutrients-14-05166-t003:** Parameters and Thresholds Used for GLIM Severity Grading in the Present Study.

Grade	Phenotypic Criteria		
	Weight Loss (%)	Low BMI (kg/m^2^)	Reduced Muscle Mass ^a,b,c^
Moderate malnutrition	5–10% within the past 6 months, or 10–20% beyond 6 months	<18.5 if <70 years, or <20 if ≥70 years	Mid-arm muscle circumference < p15, weight-standardized hand grip strength < p15
Severe malnutrition	>10% within the past 6 months, or >20% beyond 6 months	<17.0 if <70 years, or <17.8 if ≥70 years	Mid-arm muscle circumference < p5, weight-standardized hand grip strength < p5

GLIM, Global Leadership Initiative on Malnutrition; BMI, body mass index; p15, 15th percentile; p5, 5th percentile; a, males and females were evaluated separately; b, requires one phenotypic criterion meeting this grade; c, percentile values of mid-arm muscle circumference (male: p15 = 18.63 cm, p5 = 16.24 cm; female: p15 = 16.71 cm, p5 = 14.47 cm); percentile values of weight-standardized hand grip strength (hand grip strength/weight, male: p15 = 0.32, p5 = 0.22; female: p15 = 0.21, p5 = 0.14).

**Table 4 nutrients-14-05166-t004:** Baseline Characteristics of the development cohort.

Characteristics	Overall	GLIM Diagnosis			*p*
		Normal	Moderate Malnutrition	Severe Malnutrition	
	(*n* = 3612)	(*n* = 1892)	(*n* = 947)	(*n* = 773)	
**General information**					
Age, years, mean ± SD	64.09 ± 12.45	63.12 ± 11.95	64.80 ± 12.63	65.58 ± 13.20	<0.001
Sex, male, *n* (%)	2173 (60.2)	1125 (59.5)	564 (59.6)	484 (62.6)	0.291
Smoking, yes, *n* (%)	1361 (37.7)	684 (36.2)	364 (38.4)	313 (40.5)	0.095
Alcohol drinker, yes, *n* (%)	680 (18.8)	348 (18.4)	181 (19.1)	151 (19.5)	0.765
Family cancer history, yes, *n* (%)	561 (15.5)	297 (15.7)	145 (15.3)	119 (15.4)	0.958
TNM Stage, *n* (%)					0.937
Ⅰ	212 (5.9)	113 (6.0)	58 (6.1)	41 (5.3)	
Ⅱ	942 (26.1)	502 (26.5)	236 (24.9)	204 (26.4)	
Ⅲ	1482 (41.0)	764 (40.4)	400 (42.2)	318 (41.1)	
Ⅳ	976 (27.0)	513 (27.1)	253 (26.7)	210 (27.2)	
Organ metastasis, *n* (%)					0.371
0	2565 (71.0)	1343 (71.0)	673 (71.1)	549 (71.0)	
1	626 (17.3)	347 (18.3)	154 (16.3)	125 (16.2)	
2	199 (5.5)	97 (5.1)	59 (6.2)	43 (5.6)	
≥3	222 (6.1)	105 (5.5)	61 (6.4)	56 (7.2)	
Differentiation grade, *n* (%)					0.017
Well	141 (3.9)	81 (4.3)	33 (3.5)	27 (3.5)	
Moderate	2791 (77.3)	1494 (79.0)	712 (75.2)	585 (75.7)	
Poor	680 (18.8)	317 (16.8)	202 (21.3)	161 (20.8)	
Radical resection, yes, *n* (%)	2331 (64.5)	1211 (64.0)	615 (64.9)	505 (65.3)	0.774
Adjuvant chemotherapy, yes, *n* (%)	1304 (36.1)	725 (38.3)	322 (34.0)	257 (33.2)	0.014
KPS score, mean ± SD	85.78 ± 13.79	88.39 ± 10.97	84.37 ± 14.69	81.11 ± 17.00	<0.001
**Nutrition-related information**					
BMI, kg/m^2^, mean ± SD	22.43 ± 3.32	23.48 ± 2.83	21.97 ± 3.26	20.45 ± 3.45	<0.001
Mid-arm muscle circumference, cm, mean ± SD	21.12 ± 3.48	21.82 ± 3.33	20.91 ± 3.03	19.69 ± 3.85	<0.001
Hand grip strength/weight ratio, mean ± SD	0.42 ± 0.15	0.43 ± 0.14	0.41 ± 0.14	0.40 ± 0.18	<0.001
Calf circumference, cm, mean ± SD	32.82 ± 4.20	33.83 ± 4.14	32.32 ± 3.68	30.96 ± 4.21	<0.001
PGSGA score, ≥4, *n* (%)	2274 (63.0)	638 (33.7)	896 (94.6)	740 (95.7)	<0.001
NRS2002 score, ≥3, *n* (%)	1204 (33.3)	169 (8.9)	539 (56.9)	496 (64.2)	<0.001
Parenteral nutritional support, yes, *n* (%)	1061 (29.4)	474 (25.1)	291 (30.7)	296 (38.3)	<0.001
Enteral nutritional support, yes, *n* (%)	1217 (33.7)	558 (29.5)	336 (35.5)	323 (41.8)	<0.001
**Laboratory findings**					
Total protein, g/L, mean ± SD	67.75 ± 8.08	68.65 ± 7.60	67.18 ± 8.38	66.23 ± 8.54	<0.001
Albumin, g/L, mean ± SD	39.20 ± 10.67	40.40 ± 13.59	38.42 ± 5.28	37.20 ± 6.16	<0.001
Prealbumin, mg/L, mean ± SD	211.46 ± 81.56	224.36 ± 75.98	205.04 ± 87.04	187.76 ± 81.67	<0.001
Direct bilirubin, μmol/L, mean ± SD	4.55 ± 9.89	4.01 ± 7.58	5.10 ± 12.91	5.21 ± 10.52	0.003
C-reactive protein, mg/L, mean ± SD	19.17 ± 35.64	15.16 ± 29.14	19.96 ± 35.11	28.00 ± 47.25	<0.001
Hemoglobin, g/L, mean ± SD	121.25 ± 23.66	125.39 ± 21.94	118.72 ± 25.54	114.23 ± 23.26	<0.001
White blood cells, 10^9^/L, mean ± SD	6.40 ± 3.34	6.20 ± 3.30	6.40 ± 3.03	6.90 ± 3.71	<0.001
Neutrophils, 10^9^/L, mean ± SD	5.63 ± 9.85	5.32 ± 9.52	5.56 ± 9.84	6.49 ± 10.60	0.021
Red blood cells, 10^12^/L, mean ± SD	4.29 ± 2.72	4.43 ± 3.57	4.16 ± 0.63	4.12 ± 1.64	0.006
Platelets, 10^9^/L, mean ± SD	224.06 ± 92.95	214.87 ± 84.34	232.58 ± 99.35	236.10 ± 102.23	<0.001

GLIM, the Global Leadership Initiative on Malnutrition; SD, standard deviation; KPS, Karnofsky performance score; BMI, body mass index; PG-SGA, the Patient-Generated Subjective Global Assessment; NRS 2002, Nutrition Risk Screening 2002.

## Data Availability

The datasets used and/or analyzed during the current study are available from the corresponding author on reasonable request.

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
