# Peer review of "Mid-Arm Muscle Circumference or Body Weight-Standardized Hand Grip Strength in the GLIM Superiorly Predicts Survival in Chinese Colorectal Cancer Patients"

_nutrients, 2022, doi:10.3390/nu14235166_

Round 1

Reviewer 1 Report

Dear Authors,

Thank you for your manuscript. The study is well-designed, and the paper is well-written. The study aims to reveal the effect of malnutrition on cumulative survival probability and predict mortality risk in CRC patients. I have only a few minor technical comments.

1. Multiple abbreviations are used throughout the text. Providing a list of abbreviations in one place, for example, at the end of the manuscript, would make it easier to follow.

2. Possibly I missed that, but the mean or median time of the follow-up should be presented in the abstract and the Methods section. Also, please provide the mean (±SD) and range age of the study participants in the abstract.

3. The comments on the hazard ratios are incomplete in the abstract and the results section with Table 2: "Patients diagnosed with severe malnutrition based on either mid-arm circumference (MAMC) or body weight-standardized hand grip strength (HGS/W) method had the highest hazard ratio (HR, 1.51; 95% CI, 1.34-1.70; P < 0.001)" - mortality hazard ratio? Also, comments should be specified in lines 178-181 and the title of Table 2 (predictive effects of different factors on an increased mortality risk are presented in CRC patients with moderate and severe malnutrition compared to normal nutritional status).

4. Also, it is recommended to move the measures from section 2.2 to a separate section and provide a more detailed description with the references (BMI, MAMC, HGS, etc.).

Author Response

Reviewer 1:

Thank you for your manuscript. The study is well-designed, and the paper is well-written. The study aims to reveal the effect of malnutrition on cumulative survival probability and predict mortality risk in CRC patients. I have only a few minor technical comments.

  1. Multiple abbreviations are used throughout the text. Providing a list of abbreviations in one place, for example, at the end of the manuscript, would make it easier to follow.

Response: Thank you for your careful work and valuable advice. We have made a list of abbreviations and provided it at the end of the manuscript (page 18).

  1. Possibly I missed that, but the mean or median time of the follow-up should be presented in the abstract and the Methods section. Also, please provide the mean (±SD) and range age of the study participants in the abstract.

Response: Thank you for your careful work and valuable advice. We have added the median time of the follow-up in the abstract and the Methods section, as well as the mean (±SD) age of the study participants in the abstract.

[page 1 line 20-23, “During a median follow-up period of 4.2 (4.0, 4.4) years, a development cohort of 3612 CRC patients with a mean age of 64.09 ± 12.45 years was observed, as well as an external validation cohort of 875 CRC patients.”

page 2 line 88, “The median time of the follow-up was 1540 (1478, 1624) day.”]

  1. The comments on the hazard ratios are incomplete in the abstract and the results section with Table 2: "Patients diagnosed with severe malnutrition based on either mid-arm circumference (MAMC) or body weight-standardized hand grip strength (HGS/W) method had the highest hazard ratio (HR, 1.51; 95% CI, 1.34-1.70; P < 0.001)" - mortality hazard ratio? Also, comments should be specified in lines 178-181 and the title of Table 2 (predictive effects of different factors on an increased mortality risk are presented in CRC patients with moderate and severe malnutrition compared to normal nutritional status).

Response: Thank you for your careful work and valuable advice. Your understanding is absolutely correct. In our study, hazard ratio means mortality hazard ratio. We regret not making this clear in the manuscript. We have revised them to more clear descriptions. [page 1 line 29-31, “Patients diagnosed with severe malnutrition based on either mid-arm muscle circumference (MAMC) or body weight-standardized hand grip strength (HGS/W) method had the highest mortality hazard ratio (HR, 1.51; 95% CI, 1.34-1.70; P < 0.001). GLIM-defined malnutrition was diagnosed in 47.6% of patients. Severe malnutrition was an independent mortality risk factor for OS (HR, 1.25; 95% CI, 1.10-1.42; P < 0.001); page 4 line 190, “Simultaneously, patients with severe malnutrition diagnosed using MAMC or HGS/W method had the highest mortality hazard ratio (HR) compared with patients in the normal group (HR, 1.51; 95% CI, 1.34-1.70; P < 0.001).”]

Besides, we added comments on the results of Table 2 in accordance with your suggestion. [page 5 line 194-197, “Moreover, the results of table 2 indicated that predictive effects of different RMM assessment approach on an increased mortality risk are presented in CRC patients with moderate and severe malnutrition compared to normal nutritional status.”]

  1. Also, it is recommended to move the measures from section 2.2 to a separate section and provide a more detailed description with the references (BMI, MAMC, HGS, etc.).

Response: Thank you for your careful work and valuable advice. We have moved the measures from section 2.2 to section 2.3 and provided a more detailed description with the references.

[page 3 line 107-119, “2.3 Anthropometric measurements The project-trained nutritionist or clinician should take the measurement in person, avoid asking the patient and fill it in directly. Height and weight measurement require an empty stomach, no shoes, and single clothes. Body mass index (BMI) was calculated by dividing height (m) by weight (kg) squared. The mid-arm circumference (MAC) and triceps skinfold thickness (TSF) should be examined with a non-dominant arm, usually the left hand, and the patient's arms naturally droop. MAMC was calculated using the formula: MAC (cm) − 3.14 × TSF (cm). The hand grip strength (HGS) was measured with a non-dominant arm, and the average value was obtained by measuring it three times. The body weight-standardized HGS (HGS/W) was calculated by dividing HGS by weight. When measuring the calf circumference (CC), the patient was required to lie on his back. The left leg would be measured for 3 consecutive times and the maximum value would be taken.”]

Reviewer 2 Report

-We cannot agree with the statement expressed by the authors that "malignant tumors located in the colorectal area, which directly damage the digestive tract of patients, with consequent reduction in food intake and assimilation, are more prone to malnutrition compared to other non-digestive cancers”. This can be significant for cancers of the esophagus and stomach, not for colorectal cancers.

The rectum represents the most frequent location of colorectal tumors and due to its function as a reservoir it does not play a role in the assimilation of food. The same can be said for sigmoid, descending, transverse, cecum localizations, since the large intestine has the main function in water reabsorption.

As highlighted by the authors in table 4, malnutrition increases significantly with the spread of the disease, in the advanced stages but is not determined by the primary location of the tumor.

-The authors should explain why the Global Leadership Initiative on Malnutrition (GLIM) born only in 2016 was preferred over the Bioelectric Impedance Analysis (BIA) which dates to 1962.

The study, however, is valuable and deserves to be developed for clinical utility, for example, with the characteristic between malnutrition and early postoperative complications (surgical site infection, colorectal anastomosis dehiscence) and malnutrition and prolonged hospital stays.

Author Response

Reviewer 2:

-We cannot agree with the statement expressed by the authors that "malignant tumors located in the colorectal area, which directly damage the digestive tract of patients, with consequent reduction in food intake and assimilation, are more prone to malnutrition compared to other non-digestive cancers”. This can be significant for cancers of the esophagus and stomach, not for colorectal cancers.

The rectum represents the most frequent location of colorectal tumors and due to its function as a reservoir it does not play a role in the assimilation of food. The same can be said for sigmoid, descending, transverse, cecum localizations, since the large intestine has the main function in water reabsorption.

As highlighted by the authors in table 4, malnutrition increases significantly with the spread of the disease, in the advanced stages but is not determined by the primary location of the tumor.

Response: Thank you for your careful work and valuable advice. We are really sorry for misunderstanding the role of the colon and rectum in food digestion and absorption. After reading your detailed explanation, we consulted the data and clearly know that the absorption of nutrients is mainly completed in the small intestine and the rectum mainly reabsorbs water. We fully agree with your suggestion that malignant tumors located in the colorectal area are no more prone to malnutrition than other non-digestive cancers. We have removed such a statement from the manuscript. Thank you again for your detailed and professional explanation.

-The authors should explain why the Global Leadership Initiative on Malnutrition (GLIM) born only in 2016 was preferred over the Bioelectric Impedance Analysis (BIA) which dates to 1962. 

Response: Thank you for your careful work and valuable advice. First, GLIM is a two-step approach for malnutrition diagnosis and classification based on three phenotypic and two etiologic criteria. Three phenotypic criteria consist of non-volitional weight loss, low body mass index, and reduced muscle mass. Two etiologic criteria include reduced food intake or assimilation and inflammation or disease burden. From the above three phenotypic and two etiologic criteria, the patient's nutrition status can be comprehensively assessed and classified. Therefore, GLIM is a prevalent method that has been studied widely. Second, bioelectrical impedance analysis (BIA) can be used to measure body composition, such as the percentage of body fat and fat-free body mass index. However, only body composition data can be obtained through BIA; it is not comprehensive enough to assess the nutrition status without evaluation of the patient's disease condition. Therefore, GLIM was preferred over the BIA. According to your advice, we have added the explanation in the discussion section.

[page 13 line 401-408, “The application of bioelectrical impedance analysis (BIA) to measure body composition is an analytical technique developed in the last 20 years. BIA can be used to measure body composition, such as the percentage of body fat, fat-free body mass index. However, only body composition data can be obtained through BIA; it is not comprehensive enough to assess the nutrition status without evaluation of the patient's disease condition. With three phenotypic and two etiologic criteria, GLIM can comprehensively assess patients’ nutrition status and classify the malnutrition severity. Therefore, GLIM might be preferred over the BIA in malnutrition diagnosis.”]

The study, however, is valuable and deserves to be developed for clinical utility, for example, with the characteristic between malnutrition and early postoperative complications (surgical site infection, colorectal anastomosis dehiscence) and malnutrition and prolonged hospital stays.

Response: Thank you for your careful work and valuable advice. We totally agree with your viewpoint. Malnutrition reduces the tolerability and effectiveness of various cancer treatments and is a powerful predictor of postoperative complications, long-term hospitalization, readmission, and mortality in patients. Thus, the validation of GLIM is necessary and significant for assessing malnutrition in hospitalized patients.

Reviewer 3 Report

The authors present interesting data regarding the impact of mid-arm circumference and body-weight standard hand grip strength for the diagnosis of malnutrition according to GLIM criteria in order to predict a 1.25 times higher mortality risk in Chinese patients with colorectal cancer (CRC). The methodology included a big cohort of 3612 CRC patients with an external validation of 875 patients. The paper is well written.

Assuming that CT scan was available in  most of the cancer patients the question arises why this data has no been exploited for the determination of skeletal  muscle index and a comparison with the more “traditional” MAC and HGS. Furthermore, the CRP/ Albumine ratio as a validated parameter may be explored in the Chinese population. These limitations should be explained and discussed more in detail.   

Author Response

Reviewer 3:

The authors present interesting data regarding the impact of mid-arm circumference and body-weight standard hand grip strength for the diagnosis of malnutrition according to GLIM criteria in order to predict a 1.25 times higher mortality risk in Chinese patients with colorectal cancer (CRC). The methodology included a big cohort of 3612 CRC patients with an external validation of 875 patients. The paper is well written.

Assuming that CT scan was available in most of the cancer patients the question arises why this data has no been exploited for the determination of skeletal muscle index and a comparison with the more “traditional” MAC and HGS. Furthermore, the CRP/ Albumin ratio as a validated parameter may be explored in the Chinese population. These limitations should be explained and discussed more in detail.   

Response: Thank you for your careful work and valuable advice. CT is indeed a better method for assessing muscle mass, and it is the gold standard at present. Cross-sectional CT images of the third lumbar spine can be used to accurately assess the muscle mass of the patient and diagnose sarcopenia. However, CT scans require medical professionals and are expensive, so it is not universally available. In our cohort, Cross-sectional CT image of the third lumbar spine was not performed, and this data was lacking, so we used physical examination indicators to evaluate muscle mass and explore the best combination to evaluate reduced muscle mass in the GLIM criteria. According to your advice, we have made a detailed explanation in the discussion section. Besides, we will consider adding CT scans to assess the patient's muscle mass in the future in our cohort, then compare CT and physical indicators to further explore which method is more standard in determining muscle loss.

[page 13-14 line 416-424, “Second, RMM was assessed based on anthropometry rather than CT scans, which might lead to a fraction of error classifications. To our knowledge, Cross-sectional CT image of the third lumbar spine is the gold standard for assessing patients’ muscle mass and diagnosing sarcopenia. However, CT scans require medical professionals and are expensive, so it is not universally available. In our cohort, Cross-sectional CT images of the third lumbar spine were not performed, and the relevant data was not acquired, so we used anthropometric measurement parameters to evaluate muscle mass. Next, we will consider adding CT scans to assess the patients’ muscle mass, and conduct a further study to compare the effect of CT and anthropometry indicators on assessing muscle mass.”]

    When we selected independent predictors for constructing a nomogram, CRP and albumin were considered. However, according to the results of the LASSO regression analysis, CRP and albumin were eliminated. One of the reasons is that too many variables (62) were considered in the initial screening, and the effect of CRP and albumin on the survival of CRC patients might be weakened. Another reason might be incomplete data for CRP.

    C-reactive protein, as an inflammatory marker, reflects the inflammation level of the patient's body. Albumin is a main kind of protein in the body, reflecting patient’s nutritional status. As you mentioned, CRP/Albumin ratio has been validated to be an effective prognostic predictor in a large body of studies [1,2]. Relevant research was also conducted in the INSCOC cohort. Yu et al. [3] indicated that CRP could be used as a prognostic maker in malignant tumor patients. Liu et al. [4] showed that serum albumin and total protein might predict 1-year survival. However, the association between the CRP/Albumin ratio and the prognosis of cancer patients has yet to be explored with the INSCOC cohort. In the present study, we only consider the independent effect of CRP and albumin in the screening process, while the combined effect of the two variables was ignored. Thank you again for enlightening us with such an essential and necessary idea. Next, we will especially focus on CRP/Albumin ratio and validate its performance in predicting overall survival in Chinese cancer patients in our INSCOC cohort. According to your advice, we have discussed this limitation in the discussion section.

[page 14 line 424-429, “we only consider the independent effect of CRP and albumin in the screening process, while the combined effect of two variables was ignored. CRP/Albumin ratio has been indicated to be an effective prognostic predictor in previous studies. Future studies focusing on the performance of CRP/Albumin ratio in predicting overall survival in Chinese cancer patients are warranted.”]

Reference:

[1] Arakawa Y, Miyazaki K, Yoshikawa M, et al. Value of the CRP-albumin ratio in patients with resectable pancreatic cancer. J Med Invest. 2021;68(3.4):244-255. doi:10.2152/jmi.68.244

[2] Ayrancı MK, Küçükceran K, Dundar ZD. NLR and CRP to albumin ratio as a predictor of in-hospital mortality in the geriatric ED patients. Am J Emerg Med. 2021;44:50-55. doi:10.1016/j.ajem.2021.01.053

[3] Yu JM, Yang M, Xu HX, et al. Association Between Serum C-Reactive Protein Concentration and Nutritional Status of Malignant Tumor Patients. Nutr Cancer. 2019;71(2):240-245. doi:10.1080/01635581.2018.1524019

[4] Liu XY, Zhang X, Ruan GT, et al. One-Year Mortality in Patients with Cancer Cachexia: Association with Albumin and Total Protein. Cancer Manag Res. 2021;13:6775-6783. Published 2021 Aug 29. doi:10.2147/CMAR.S318728

Round 2

Reviewer 2 Report

The authors addressed the comments of the reviewer